# National assessment on the frequency of pain medication prescribed for intrauterine device insertion procedures within the Veterans Affairs Health Care System

**Anna D. Ware**[1]*, **Terri L. Blumke**[1], **Peter J. Hoover**[1], **Zach P. Veigulis**[2], **Jacqueline M. Ferguson**[3], **Malvika Pillai**[1,4], **Thomas F. Osborne**[1,5]

**1** US Department of Veterans Affairs, Palo Alto Healthcare System, National Center for Collaborative Healthcare Innovation, Palo Alto, California, United States of America, **2** Department of Business Analytics, University of Iowa Tippie College of Business, Iowa City, IA, United States of America, **3** Center for Innovation to Implementation, Veterans Affairs Palo Alto Health Care System, Menlo Park, CA, United States of America, **4** Center for Biomedical Informatics Research, Stanford University School of Medicine, Stanford, CA, United States of America, **5** Department of Radiology, Stanford University School of Medicine, Stanford, CA, United States of America

* denee.ware@outlook.com

**Data Availability Statement:** Due to the sensitive nature of the data used in this assessment, which

## Abstract

### Background

The intrauterine device (IUD) is a highly effective form of long-acting reversible contraception, widely recognized for its convenience and efficacy. Despite its benefits, many patients report moderate to severe pain during and after their IUD insertion procedure. Furthermore, reports suggest significant variability in pain control medications, including no adequate pain medication. The aim of this evaluation was to assess the pharmaceutical pain medication types, proportions, and trends related to IUD insertion procedures within the Veterans Health Administration (VHA).

### Methods

IUD insertion procedures documented in the VA electronic health record were assessed from 1/1/2018 to 10/13/2023. Descriptive statistics described patient and facility characteristics while annual trends were assessed using linear regression.

### Results

Out of the 28,717 procedures captured, only 11.4% had any form of prescribed pain medication identified. Non-Steroidal Anti-Inflammatory Drugs (NSAIDs) were the most frequently prescribed pain medication category (8.3%), with ibuprofen being the most common pain medication overall (6.1%). Over the assessment period, there was an average annual increase of 0.52% (*p* = 0.038) of procedures with prescribed pain medication, increasing from 10.3% in 2018 to 13.3% in 2023.

includes patient health information, and the US Department of Veterans Affairs (VA) regulations and ethics agreements, the data utilized for this assessment are not permitted to leave the VA firewall without proper authorization. Access to VA data is restricted and requires a Data Use Agreement (DUA). Researchers interested in accessing this data must first obtain approval from a relevant institutional Review Board (IRB) and have a VA-authorized study protocol. For inquiries regarding data access, researchers should initially contact the VA Palo Alto Health Care System Research & Development Committee (R&D) for ethical and regulatory review. The R&D Committee will coordinate the process, and, upon meeting the required criteria, may work with the National Center for Collaborative Healthcare Innovation (NCCHI) to facilitate data access. For more general information, please visit https://www.virec. research.va.gov or contact the VA Information Resource Center at VIReC@va.gov.

**Funding:** The author(s) received no specific funding for this work.

**Competing interests:** The authors have declared that no competing interests exist.

## Conclusions

Although IUD insertion procedures have been seeing an increase in prescribed pain medication, the overall proportion remains disproportionality low relative to the pain experienced. Additionally, when pain interventions were initiated, they disproportionally utilized medication that have been shown to be ineffective. The intent of the work is that the information will help guide data driven pain medication strategies for patients undergoing IUD insertion procedures within the VHA.

## Introduction

Intrauterine devices (IUDs) are a highly recommended long-acting reversible contraception (LARC) option not only due to their mean one-year failure rate of less than 1%, but also because many patients prefer them for their convenience, minimal maintenance, and long-term effectiveness [1, 2] For some individuals, IUDs may be the preferable or only option for contraception due to specific health conditions. Those who cannot tolerate hormonal contraception, such as individuals with migraines, hypertension, or those at increased risk of thromboembolism, may opt for hormone-free copper IUDs [2]. Additionally, hormonal IUDs also provide non-contraceptive benefits, such as reducing heavy menstrual bleeding and dysmenorrhea [1, 2].

However, pain surrounding the insertion of the IUD has been gaining widespread attention due to numerous negative patients' experiences [3–6]. Although patient-reported pain for this procedure can vary [3, 6], a recent survey reported that 78% of respondents rated their IUD insertion pain as moderate to severe, with 46% experiencing vasovagal symptoms [5]. In attempts to mitigate pain during IUD insertion, several pharmacological interventions have been tested with mixed results [6–13]. For example, certain formulations of lidocaine, such as cervical lidocaine gel and paracervical blocks, have been shown to be effective in reducing pain both during and after the procedure [6–9, 11, 13]. In contrast, most non-steroidal anti-inflammatory drugs (NSAIDs), including ibuprofen and ketorolac, have generally been ineffective at managing insertion-related pain, while naproxen has demonstrated some promise [6–8, 10]. Cervical ripening agents (prostaglandins), like misoprostol, were once thought to ease the insertion process, but recent studies have cast doubt on their effectiveness [6–8, 12] Additionally, while opioids are not commonly prescribed for this procedure, tramadol has been shown to reduce pain during IUD insertion in some studies [9–11].

During the procedure, the IUD is inserted by clamping the cervix using a tenaculum, followed by measuring the depth of the uterine cavity, placement of the IUD, trimming of the strings, and confirmation of appropriate placement [14]. While some healthcare providers refer to the sensation caused by the cervical clamping as a 'quick pinch', many patients have reported severe pain [5, 15]. The cervical clamping is often then followed by menstrual-pain-like cramps as the IUD is inserted into the uterus [16]. For some, the pain of insertion lasts considerably longer than the approximately 15-minute procedure [16]. As a result, many patients have called for a comprehensive strategy to manage pain associated with IUD insertion within the United States (US) [15, 17–20]. Notably, the Center for Disease Control and Prevention (CDC) updated its guidelines in 2024 asking providers to discuss effective pain management strategies with their patients prior to IUD placement [21].

The Veterans Health Administration (VHA) is the largest integrated healthcare system in the US, serving over 9 million enrolled Veterans annually [22]. Women currently represent approximately 10% of enrolled Veterans, a figure that is expected to rise to 18% by 2040 [23,

24]. Recently, the White House released an executive order to announce new actions to advance health research and innovation for women and gender-diverse individuals to "improve women's lives across America." [25] To meet the needs of this rapidly expanding population, VHA has focused on key initiatives to enhance access to comprehensive women's healthcare through specifically trained providers [26, 27]. Veterans can now access a full range of contraceptive options with no or minimal out-of-pocket expenses as part of their comprehensive primary care [28]. This enhanced accessibility, coupled with the availability of specifically trained providers in VA women's health clinics, enables Veterans to access LARCs more readily than the general population. Consequently, LARC utilization rates, including IUDs and implants, have been reported to be higher among Veterans than in the general population. According to a national telephone survey, approximately 23% of Veterans assigned female at birth (AFAB) of childbearing age (18–45 years) at risk for unintended pregnancy utilize a LARC, compared to only 11% of the general US population. This survey also found that 65% of these Veterans received their IUD within a VA outpatient clinic [29].

Yet, despite the growing numbers of women Veterans within VHA and the prevalence of LARC use among Veterans, it is unknown how frequently pain medication is prescribed for IUD insertion procedures. In this assessment, we examined the type and extent to which pain medication is prescribed for IUD insertions performed in an outpatient setting among Veterans receiving care within the VHA. The intent is that this information and knowledge will help guide data driven quality improvement efforts within the VHA.

## Materials and methods

### Data sources

We performed a retrospective assessment of electronic health record (EHR) data for patients who were AFAB, aged 18 years or older, and were receiving care at VA medical centers. The assessment focused on those who underwent an IUD insertion procedure within an outpatient clinic or outpatient surgical visit at a VHA facility from January 1st, 2018, to October 13th, 2023. Procedures were captured using the CPT code 58300: "Insertion of IUD." [30]. Outpatient visits where other procedures were performed other than just the IUD insertion (for example, the visit included CPT codes 58300: "Insertion of IUD" and 56740: "Excision of Bartholin's gland or cyst") were excluded from our assessment as it could not be determined if prescribed pain interventions were intended for the purpose of the IUD insertion or the other procedures present. All data was queried from the VA's Corporate Data Warehouse (CDW), which is a relational database that aggregates EHR data from all VHA facilities [31]. Prescribed non-opioid analgesic pain interventions were queried from outpatient medication tables at the IUD consult or within (-) 45 days, (+) 1 day surrounding the procedure, as preprocedural consults typically occur within 45 days prior to procedure. Any prescribed opioid analgesics, a highly monitored drug class within VHA, were captured within (+/-) 24 hours of the procedure. Lastly, leveraging SQL string search techniques, we parsed nursing text orders and patient discharge instructions within procedure encounter notes to capture any medications not already documented in structured data within (+/-) 24 hours surrounding the procedure. This allowed the inclusion of pain medications prescribed and recorded through both structured and unstructured EHR data.

### Pharmacological IUD pain medication interventions

Prescribed pharmacological pain interventions associated with the IUD insertion procedure were grouped into five categories, as documented elsewhere [6, 7]: (1) NSAIDs, (2) Lidocaine,

(3) Prostaglandins, (4) Opioid Analgesics, and (5) Combination or Other. List of drugs included in each category are available in **S1 Appendix**.

## Co-variates

We included a range of patient sociodemographic and clinical characteristics to account for potential disparities in care and factors that may influence access to care. Sociodemographic variables such as age, self-identified race or ethnicity, marital status, third-party (non-VA) insurance, and rurality of patient's residence were included to capture differences in healthcare access and utilization [32, 33]. In addition, clinical characteristics like body mass index (BMI), parity status (nulliparous vs. parous), Charlson Comorbidity Index (CCI) score, and female-specific comorbidities (e.g., chronic pelvic pain, postpartum depression, dyspareunia, history of military sexual trauma (MST)) were included to identify factors that may influence an individual's medical need for pain management [34].

We also captured provider and facility characteristics including the 4 US regions (as defined by the Centers for Disease Control, 2023) [35], facility complexity (VHA classification system for VA medical centers that ranges from high, medium, and low) [36, 37], state location of the clinic, outpatient care setting (e.g., gynecology clinic, primary care clinic), and the type of primary provider (e.g., gynecologist, nurse practitioner) associated with the IUD insertion procedure to assess variability in prescribing practices. Facility complexity was included to account for differences in resources availability, which may influence pain management practices. High complexity facilities have larger levels of patient volume, higher patient risk, more teaching and research resources, and contain level 4 to 5 intensive care units compared to low complexity facilities which have little or no teaching/research, the lowest number of physician specialists per patient, and contain level 1 and 2 intensive care units [36, 37].

## Statistical analysis

The data collected for this assessment was analyzed from October 14[th], 2023, to July 17[th], 2024. All data was fully anonymized before accessing the data. Descriptive statistics to summarize patient, provider, and facility characteristics and compared differences in the prevalence of these characteristics among those prescribed pain medication compared to those without. Differences between groups were evaluated using Chi-square and the student's t-tests. To quantitatively assess any temporal trends, we applied a linear regression model to evaluate any changes in the number of IUD insertions with prescribed pain medications by year. Data were analyzed using R Statistical Software (v4.1.2; R Core Team 2021). This quality assessment project received determination of non-research from Stanford Institutional Review Board, (Stanford University, Stanford, CA, USA), which waived the need for participant consent.

## Results

### National VHA results

Among the 1,614,650 patients who were AFAB, aged 18 years or older, and who received care within the VHA during the assessment period, 28,717 IUD insertion procedures were performed across VHA, nationally (flowchart available in **S2 Appendix**). Out of the 28,717 procedures captured, 11.4% (3,260) had any form of pharmaceutical pain management prescribed (**Table 1**). When analyzing annual trends, the slope of the fitted linear regression model indicated that, on average, there was a statistically significant annual increase of 0.52% ($p = 0.038$; **Fig 1**) in the percentage of procedures with prescribed pain medication, increasing from 10.3% in 2018 to 13.3% in 2023.

**Table 1. Patient characteristics among the 24,010 IUD insertions performed within VHA between January 1st, 2018, to October 13th, 2023, with any documented pain medication for their intrauterine device insertion procedure compared to those without any prescribed pain medication.**

| | Total (Column frequencies) | No Pain Medication Prescribed | Pain Medication Prescribed | P-value [a] |
|---|---|---|---|---|
| **Total, n (%)** | **28,717 (100.0%)** | **25,457 (88.7%)** | **3,260 (11.4%)** | |
| *Sociodemographic characteristics* | | | | |
| **Age (years), mean (SD)** | 39.53 (7.58) | 39.58 (7.56) | 39.08 (7.71) | 0.135 |
| **Race and/or Ethnicity, n (%)** | | | | <0.001 |
| American Indian or Alaskan Native | 412 (1.4%) | 378 (91.7%) | 34 (8.3%) | |
| Asian | 884 (3.1%) | 773 (87.4%) | 111 (12.6%) | |
| Black or African American | 7,165 (25.0%) | 6,246 (87.2%) | 919 (12.8%) | |
| Hispanic | 3,951 (13.8%) | 3,513 (88.9%) | 438 (11.1%) | |
| Native Hawaiian or Pacific Islander | 340 (1.2%) | 295 (86.8%) | 45 (13.2%) | |
| Non-Hispanic White | 14,089 (49.1%) | 12,598 (89.4%) | 1,491 (10.6%) | |
| Declined to Answer or Unknown | 1,876 (6.5%) | 1,654 (88.2%) | 222 (11.8%) | |
| **Marital Status, n (%)** | | | | <0.001 |
| Married | 10,649 (37.1%) | 9,574 (89.9%) | 1,075 (10.1%) | |
| Single | 8,620 (30.0%) | 7,427 (86.2%) | 1,193 (13.8%) | |
| Divorced/Separated | 8,780 (30.6%) | 7,861 (89.5%) | 919 (10.5%) | |
| Widowed | 143 (0.5%) | 127 (88.8%) | 16 (11.2%) | |
| Unknown | 525 (1.8%) | 468 (89.1%) | 57 (10.9%) | |
| **Has third party (non-VA) insurance, n (%)** | | | | 0.996 |
| No | 19,351 (67.4%) | 17,153 (88.6%) | 2,198 (11.4%) | |
| Yes | 9,348 (32.6%) | 8,286 (88.6%) | 1,062 (11.4%) | |
| Unknown | 18 (0.1%) | 18 (100.0%) | - | |
| **Rurality of Patient's Residence, n (%)** | | | | **0.003** |
| Urban | 22,374 (77.9%) | 19,713 (88.1%) | 2,661 (11.9%) | |
| Rural | 6,171 (21.5%) | 5,593 (90.6%) | 578 (9.4%) | |
| Highly Rural | 133 (0.5%) | 117 (88.0%) | 16 (12.0%) | |
| Unknown | 39 (0.1%) | 34 (87.2%) | 5 (12.8%) | |
| *Clinical characteristics* | | | | |
| **Body Mass Index, mean (SD)** | 31.30 (6.44) | 31.33 (6.42) | 31.11 (6.59) | 0.798 |
| **Parity Status [b], n (%)** | | | | **0.002** |
| Nulliparous | 27,114 (94.4%) | 23,997 (88.5%) | 3,117 (11.4%) | |
| Parous/Multiparous | 1,603 (5.6%) | 1,460 (91.1%) | 143 (8.9%) | |
| **Charlson Comorbidity Index, mean (SD)** | 0.57 (1.12) | 0.58 (1.13) | 0.55 (1.05) | 0.181 |
| *Diagnosed Conditions* | | | | |
| **Chronic pelvic pain, n (%)** | | | | <0.001 |
| Yes | 6,485 (22.6%) | 5,583 (86.1%) | 902 (13.9%) | |
| No | 22,232 (77.4%) | 19,874 (89.4%) | 2,358 (10.6%) | |
| **Dyspareunia, n (%)** | | | | <0.001 |
| Yes | 2,539 (8.8%) | 2,196 (86.5%) | 343 (13.5%) | |
| No | 26,178 (91.2%) | 23,261 (88.9%) | 2,917 (11.1%) | |
| **Post-menopausal, n (%)** | | | | 0.985 |
| Yes | 2,425 (8.4%) | 2,150 (88.7%) | 275 (11.3%) | |
| No | 26,292 (91.6%) | 23,307 (88.7%) | 2,985 (11.4%) | |
| **Dysmenorrhea, n (%)** | | | | <0.001 |
| Yes | 5,332 (18.6%) | 4,580 (85.9%) | 752 (14.1%) | |
| No | 23,385 (81.4%) | 20,877 (89.3%) | 2,508 (10.7%) | |
| **Anxiety Disorders [c], n (%)** | | | | **0.014** |

*(Continued)*

**Table 1.** (Continued)

| | Total (Column frequencies) | No Pain Medication Prescribed | Pain Medication Prescribed | P-value [a] |
|---|---|---|---|---|
| **Total, n (%)** | **28,717 (100.0%)** | **25,457 (88.7%)** | **3,260 (11.4%)** | |
| Yes | 18,004 (62.7%) | 15,896 (88.3%) | 2,108 (11.7%) | |
| No | 10,713 (37.3%) | 9,561 (89.3%) | 1,152 (10.8%) | |
| **Postpartum depression, n (%)** | | | | 0.703 |
| Yes | 273 (1.0%) | 244 (89.4%) | 29 (10.6%) | |
| No | 28,444 (99.0%) | 25,213 (88.6%) | 3,231 (11.4%) | |
| **History of Military Sexual Trauma, n (%)** | | | | **0.041** |
| Yes | 10,981 (38.2%) | 9,681 (88.2%) | 1,300 (11.8%) | |
| No | 17,736 (61.8%) | 15,776 (88.9%) | 1,960 (11.1%) | |

[a] Statistical significance was determined by Chi-Square or T-test. Significant values ($p$-value $< 0.05$) are bolded.

[b] Data for this variable may include instances of missingness that could not be fully accounted for in the analysis

[c] Anxiety disorders included panic disorder (episodic paroxysmal anxiety), generalized anxiety disorder, other specific anxiety disorders, anxiety disorder (unspecified), and other mixed anxiety disorders.

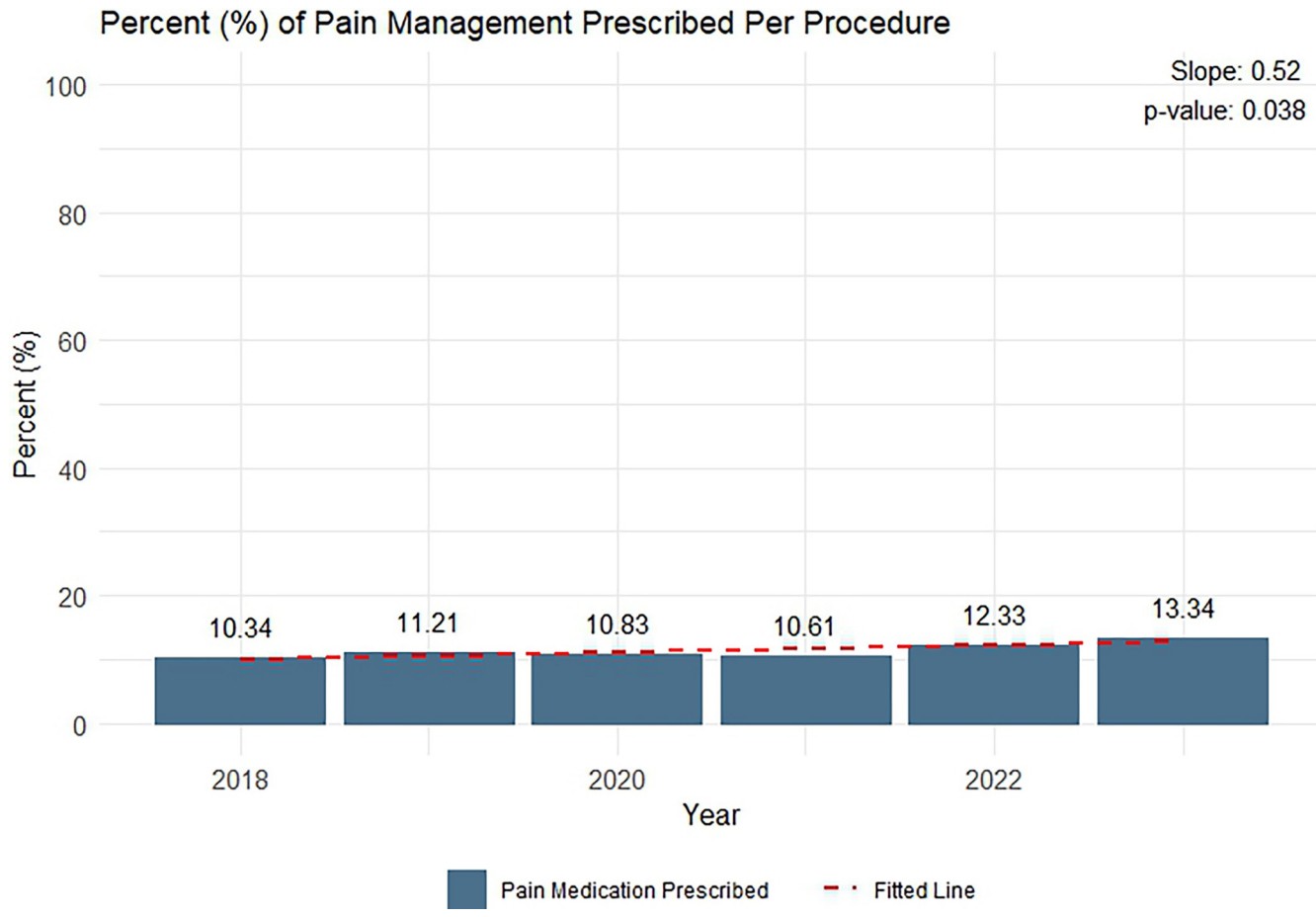

**Fig 1. Percent of IUD insertion procedures among the 24,010 IUD insertions performed within VHA between January 1st, 2018, to October 13th, 2023, with any pharmaceutical pain medication prescribed within the Veterans Affairs Healthcare System from January 1st, 2018, to October 13th, 2023. F1L:** This graph illustrates the annual percentage of intrauterine device insertion procedures with any pharmaceutical pain medication prescribed within the Veterans Affairs Healthcare System from January 1st, 2018, to October 13th, 2023. The navy bars represent the mean percentage of procedures with pharmaceutical pain medication prescribed per year. The dashed red line depicts the linear regression line fitted to the observed data. The slope of this line with a p-value indicates the statistical significance of this trend. Each observed data point is labeled with the corresponding percentage.

## Patient characteristics

We found minimal differences in patients' sociodemographic information among those who received pharmaceutical pain management for their procedure compared to those who received none. For both the pain medication and no pain medication cohorts, average age was around 39 years (SD: 7.6), and roughly 33% had third-party insurance. Slightly higher proportions of Native Hawaiian or Pacific Islanders (13.2%) and Black or African Americans (12.8%) were prescribed pain medication compared to Non-Hispanic Whites (10.6%) and American Indian or Alaskan Natives (8.3; p<0.001). For marital status, 13.8% of patients who self-identified as "Single" were prescribed pain medication while 10.1% of those self-identified as "Married" were prescribed (p<0.001). Additionally, 11.9% of urban patients were prescribed pain medication compared to only 9.4% of rural patients (p = 0.003; **Table 1**).

When analyzing history of childbirths, we found a higher proportion of nulliparous patients prescribed pain medication compared to parous/multiparous patients (11.4% vs 8.9%; p<0.001). Assessing comorbidities revealed that CCI scores were low for both groups, although slightly higher among those not prescribed pain medication (average [SD]: 0.58 [1.13] vs 0.55 [1.05]). Analysis of specific related conditions demonstrated statically significant results for those receiving pain medication and being diagnosed or not (respectively) with the following conditions: anxiety disorders (11.7% vs. 10.8%; p = 0.014), dysmenorrhea (14.1% vs. 10.7%; p<0.001), dyspareunia (13.5% vs 11.1%; p<0.001), chronic pelvic pain (13.9% vs 10.6%; p<0.001), and those with a history of MST (11.8% vs. 11.1%; p = 0.041; **Table 1**). Patient characteristics by pain medication types can be found in **S3 Appendix**.

## IUD insertion procedures and pharmaceutical pain medication types

Out of the 28,717 IUD insertion procedures captured, NSAIDs were the most commonly prescribed pharmaceutical category (8.3%; **Table 2**), followed by prostaglandins (1.6%), opioid analgesics (0.6%), combination/other (0.6%), and lidocaine (0.2%). The most common clinic location for IUD insertions were gynecology clinics (88.5%), followed by Comprehensive Women's and Gender Diverse Primary Care Clinic (8.5%). The majority of procedures were performed at a high complexity facility (91.7%) compared to low complexity facilities (3.4%). However, there was relatively small variation in procedure locations where no pain medication was prescribed, ranging from 88.5% at high complexity facilities compared to 92.1% among procedures performed at medium complexity facilities (**Table 2**).

Procedures were most commonly performed by gynecologists (54.8%), other/non-specific physicians (20.0%), and nurse practitioners (18.0%). There was little variation among provider types in instances where no pain medication was prescribed, ranging from 87.2% among medical residents to 90.5% among nurse practitioners. There was a higher proportion of pain medication prescribed for patients receiving their IUD within the categorical area of the operating room (OR)/general surgery (22.6%), with 7.5% of these patients prescribed an NSAID and 6.0% prescribed an opioid analgesic (**Table 2**).

Among individual medications, ibuprofen was the most frequently prescribed pain intervention as a stand-alone treatment (6.1%) and was also commonly prescribed in combination with other pain intervention modalities (**Table 3**). Additionally, we found that the cervical ripening agent, misoprostol, was prescribed among 1.6% out of the total procedures performed, and ketorolac and naproxen were prescribed among 1.6% and 0.5% of procedures, respectively. Additional information on individual medications and medication combinations per clinic location can be found in **Table 3**.

**Table 2.** VA clinic region, care setting, primary provider characteristics, and medication types for intrauterine device insertion procedures performed (row frequencies, n (%)).

| | Total | Type of Pain Medication Prescribed | | | | | |
|---|---|---|---|---|---|---|---|
| | | None | NSAIDs | Combination or Other [a] | Opioid Analgesics | Prostaglandins | Lidocaine [e] |
| **Total** | **28,717** | 25,457 (88.7%) | 2,392 (8.3%) | 166 (0.6%) | 171 (0.6%) | 469 (1.6%) | 62 (0.2%) |
| *Facility Characteristics* | | | | | | | |
| **Region** | | | | | | | |
| South | **13,535 (47.1%)** | 11,973 (88.5%) | 1,208 (8.9%) | 86 (0.6%) | 36 (0.3%) | 205 (1.5%) | 27 (0.2%) |
| West | **7,397 (25.8%)** | 6,503 (87.9%) | 604 (8.2%) | 35 (0.5%) | 97 (1.3%) | 147 (2.0%) | 14 (0.2%) |
| Midwest | **5,072 (17.7%)** | 4,511 (88.9%) | 434 (8.6%) | 32 (0.3%) | 26 (0.5%) | 59 (1.2%) | 10 (0.2%) |
| Northeast | **2,713 (9.5%)** | 2,470 (91.0%) | 146 (5.4%) | 13 (0.5%) | 15 (0.6%) | 58 (2.1%) | 11 (0.4%) |
| **Facility Complexity [b]** | | | | | | | |
| High complexity | **26,342 (91.7%)** | 23,314 (88.5%) | 2,248 (8.5%) | 159 (0.6%) | 162 (0.6%) | 401 (1.5%) | 58 (0.2%) |
| Medium complexity | **1,391 (4.8%)** | 1,281 (92.1%) | 56 (4.0%) | 5 (0.4%) | 4 (0.3%) | 42 (3.0%) | 3 (0.2%) |
| Low complexity | **984 (3.4%)** | 862 (87.6%) | 88 (8.9%) | 2 (0.2%) | 5 (0.5%) | 26 (2.6%) | 1 (0.1%) |
| **Care Setting** | | | | | | | |
| Gynecology Clinic | **25,412 (88.5%)** | 22,570 (88.8%) | 2,108 (8.3%) | 133 (0.5%) | 138 (0.5%) | 415 (1.6%) | 48 (0.2%) |
| Comprehensive Women's and Gender Diverse Primary Care Clinic | **2,434 (8.5%)** | 2,123 (87.2%) | 232 (9.5%) | 8 (0.3%) | 10 (0.4%) | 47 (1.9%) | 14 (0.6%) |
| OR/General Surgery | **319 (1.1%)** | 247 (77.4%) | 24 (7.5%) | 23 (7.2%) | 19 (6.0%) | 6 (1.9%) | 0 (0.0%) |
| Primary Care Clinic | **552 (1.9%)** | 517 (93.7%) | 28 (5.1%) | 2 (0.4%) | 4 (0.7%) | 1 (0.2%) | 0 (0.0%) |
| *Procedure characteristics* | | | | | | | |
| **Provider Type for Procedure** | | | | | | | |
| Gynecologist | **15,741 (54.8%)** | 13,875 (88.2%) | 1,424 (9.1%) | 84 (0.5%) | 56 (0.4%) | 266 (1.7%) | 36 (0.2%) |
| Other/Non-specific Physician | **5,729 (20.0%)** | 5,104 (89.1%) | 396 (6.1%) | 49 (0.9%) | 97 (1.7%) | 70 (1.2%) | 13 (0.2%) |
| Nurse Practitioner | **5,177 (18.0%)** | 4,683 (90.5%) | 373 (7.2%) | 13 (0.3%) | 4 (0.1%) | 93 (1.8%) | 11 (0.2%) |
| RN/APRN | **501 (1.7%)** | 446 (89.0%) | 47 (9.4%) | 4 (0.8%) | 1 (0.2) | 3 (0.6%) | 0 (0.0%) |
| Surgeon | **402 (1.7%)** | 320 (79.6%) | 56 (13.9%) | 11 (2.7%) | 11 (2.7%) | 3 (0.8%) | 1 (0.3%) |
| Medical Resident [c] | **164 (0.6%)** | 143 (87.2%) | 16 (9.8%) | 1 (0.6%) | 0 (0.0%) | 4 (2.4%) | 0 (0.0%) |
| Other Provider Type [d] | **211 (0.7%)** | 183 (86.7%) | 24 (11.4%) | 0 (0.0%) | 1 (0.5%) | 2 (1.0%) | 1 (0.5%) |

NSAIDs = Non-Steroidal Anti-inflammatory Drugs; WH = Women's Health; OR = Operating Room; RN = Registered Nurse; APRN = Advanced Practice Registered Nurse

[a] Combination or Other included combination NSAID +/- Lidocaine, Opioid Analgesics, and/or Misoprostol, or General anesthesia +/- other pharmacological interventions.

[b] Station complexity: high complexity facilities have large levels of patient volume, patient risk, teaching and research, and contain level 4 to 5 intensive care units; medium complexity facilities have medium levels of patient volume, medium patient risk, some teaching and/or research, and contain level 3 and 4 intensive care units; low complexity facilities have the smallest level of patient volume, little or no teaching/research, the lowest number of physician specialists per patient, and contain level 1 and 2 intensive care units.

[c] Procedure signed off by medical resident, while being supervised by a VA physician.

[d] Other Provider Type includes Physician's Assistant, Licensed Practical Nurse/Licensed Vocational Nurse, and Health Technician.

[e] Lidocaine includes diclofenac (plus lidocaine), cervical lidocaine gel, the paracervical block, and cervical lidocaine-prilocaine cream

**Table 3. Individual medications and medication combinations prescribed for intrauterine device insertion procedure by VA care setting.**

| Medication Group | Medications or Medication Combinations | Overall (N = 28717) | Gynecology Clinic (n = 25412) | OR/ General Surgery (n = 319) | Primary Care Clinic (n = 552) | Comprehensive Women's and Gender Diverse Primary Care Clinic (n = 2434) |
|---|---|---|---|---|---|---|
| None | None | 25457 (88.6%) | 22570 (88.8%) | 247 (77.4%) | 517 (93.7%) | 2123 (87.2%) |
| NSAIDs[1] | | | | | | |
| | Ibuprofen | 1740 (6.1%) | 1520 (6.0%) | 19 (6.0%) | 23 (4.2%) | 178 (7.3%) |
| | Ketorolac | 460 (1.6%) | 409 (1.6%) | 1 (0.3%) | 4 (0.7%) | 46 (1.9%) |
| | Naproxen | 132 (0.5%) | 123 (0.5%) | 2 (0.6%) | 1 (0.2%) | 6 (0.2%) |
| | Ketorolac + Ibuprofen | 49 (0.2%) | 47 (0.2%) | 2 (0.6%) | - | - |
| | Naproxen + Ibuprofen | 8 (0.0%) | 6 (0.0%) | - | - | 2 (0.1%) |
| | Ketorolac + Naproxen | 3 (0.0%) | 3 (0.0%) | - | - | - |
| Opioid Analgesics[2] | | | | | | |
| | Hydromorphone | 41 (0.1%) | 38 (0.1%) | 3 (0.9%) | - | - |
| | Fentanyl | 33 (0.1%) | 30 (0.1%) | 3 (0.9%) | - | - |
| | Acetaminophen/Hydrocodone | 22 (0.1%) | 19 (0.1%) | 2 (0.6%) | - | 1 (0.0%) |
| | Oxycodone | 20 (0.2%) | 11 (0.0%) | 5 (1.6%) | - | 4 (0.2%) |
| | Tramadol | 14 (0.0%) | 11 (0.0%) | - | 1 (0.2%) | 2 (0.1%) |
| | Fentanyl + Hydromorphone | 11 (0.0%) | 8 (0.0%) | 1 (0.3%) | 2 (0.4%) | - |
| | Codeine | 10 (0.0%) | 7 (0.0%) | - | - | 3 (0.1%) |
| | Fentanyl + Oxycodone | 9 (0.0%) | 6 (0.0%) | 2 (0.6%) | 1 (0.2%) | - |
| | Acetaminophen/Hydrocodone + Hydromorphone | 4 (0.0%) | 3 (0.0%) | 1 (0.3%) | - | - |
| | Acetaminophen/Oxycodone | 2 (0.0%) | 2 (0.0%) | - | - | - |
| | Acetaminophen/ Oxycodone + Fentanyl | 1 (0.0%) | | 1 (0.3%) | - | - |
| | Buprenorphine/Naloxone | 1 (0.0%) | 1 (0.0%) | - | - | - |
| | Hydromorphone + Oxycodone | 1 (0.0%) | 1 (0.0%) | - | - | - |
| | Morphine | 1 (0.0%) | 1 (0.0%) | - | - | - |
| | Oxycodone + Morphine | 1 (0.0%) | - | 1 (0.3%) | - | - |
| Prostaglandins[3] | | | | | | |
| | Misoprostol | 469 (1.6%) | 415 (1.6%) | 6 (1.9%) | 1 (0.2%) | 47 (1.9%) |
| Lidocaine[4] | | | | | | |
| | Diclofenac (plus Lidocaine) | 35 (0.1%) | 25 (0.1%) | - | - | 10 (0.4%) |
| | Cervical Lidocaine Gel | 14 (0.0%) | 11 (0.0%) | - | - | 3 (0.1%) |
| | Cervical LP Cream | 7 (0.0%) | 6 (0.0%) | - | - | 1 (0.0%) |
| | Paracervical Block | 6 (0.0%) | 6 (0.0%) | - | - | - |
| Combination or Other[5] | | | | | | |
| | Misoprostol + Ibuprofen | 37 (0.1%) | 35 (0.1%) | 1 (0.3%) | - | 1 (0.0%) |
| | Ketorolac + Misoprostol | 21 (0.1%) | 21 (0.1%) | - | - | - |
| | Acetaminophen/Hydrocodone + Ibuprofen | 9 (0.0%) | 7 (0.0%) | 1 (0.3%) | - | 1 (0.0%) |
| | Oxycodone + Ibuprofen | 6 (0.0%) | 3 (0.0%) | 3 (0.9%) | - | - |
| | Cervical Lidocaine Gel + Ibuprofen | 5 (0.0%) | 3 (0.0%) | - | - | 2 (0.1%) |
| | Codeine + Ibuprofen | 4 (0.0%) | 2 (0.0%) | 1 (0.3%) | 1 (0.2%) | - |
| | Acetaminophen/Hydrocodone + Ketorolac | 3 (0.0%) | 3 (0.0%) | - | - | - |

(*Continued*)

**Table 3.** (Continued)

| Medication Group | Medications or Medication Combinations | Overall (N = 28717) | Gynecology Clinic (n = 25412) | OR/ General Surgery (n = 319) | Primary Care Clinic (n = 552) | Comprehensive Women's and Gender Diverse Primary Care Clinic (n = 2434) |
|---|---|---|---|---|---|---|
| | Codeine + Ketorolac | 3 (0.0%) | 3 (0.0%) | - | - | - |
| | Diclofenac (plus Lidocaine) + Ibuprofen | 3 (0.0%) | 2 (0.0%) | - | - | 1 (0.0%) |
| | Fentanyl + Hydromorphone + Oxycodone | 3 (0.0%) | 3 (0.0%) | - | - | - |
| | Fentanyl + Ibuprofen | 3 (0.0%) | 2 (0.0%) | 1 (0.3%) | - | - |
| | Hydromorphone + Ibuprofen | 3 (0.0%) | 1 (0.0%) | 1 (0.3%) | - | 1 (0.0%) |
| | Hydromorphone + Ketorolac + Ibuprofen | 3 (0.0%) | 3 (0.0%) | - | - | - |
| | Hydromorphone + Oxycodone + Ibuprofen | 3 (0.0%) | - | 3 (0.9%) | - | - |
| | Naproxen + Misoprostol | 3 (0.0%) | 2 (0.0%) | - | 1 (0.2%) | - |
| | Oxycodone + Naproxen | 3 (0.0%) | 3 (0.0%) | - | - | - |
| | Acetaminophen/Hydrocodone + Ketorolac + Ibuprofen | 2 (0.0%) | 2 (0.0%) | - | - | - |
| | Hydromorphone + Ketorolac | 2 (0.0%) | 1 (0.0%) | 1 (0.3%) | - | - |
| | Hydromorphone + Misoprostol | 2 (0.0%) | 2 (0.0%) | - | - | - |
| | Hydromorphone + Misoprostol + Ibuprofen | 2 (0.0%) | 2 (0.0%) | - | - | - |
| | Hydromorphone + Oxycodone + Ketorolac | 2 (0.0%) | 1 (0.0%) | 1 (0.3%) | - | - |
| | Ketorolac + Diclofenac (plus Lidocaine) | 2 (0.0%) | 2 (0.0%) | - | - | - |
| | Ketorolac + Misoprostol + Ibuprofen | 2 (0.0%) | 2 (0.0%) | - | - | - |
| | Naproxen + Diclofenac (plus Lidocaine) | 2 (0.0%) | 1 (0.0%) | - | - | 1 (0.0%) |
| | Oxycodone + Ketorolac | 2 (0.0%) | 2 (0.0%) | - | - | - |
| | Acetaminophen/Hydrocodone + Fentanyl + Ibuprofen | 1 (0.0%) | 1 (0.0%) | - | - | - |
| | Acetaminophen/Hydrocodone + Ketorolac + Misoprostol | 1 (0.0%) | 1 (0.0%) | - | - | - |
| | Acetaminophen/Hydrocodone + Ketorolac + Misoprostol + Ibuprofen | 1 (0.0%) | 1 (0.0%) | - | - | - |
| | Acetaminophen/Hydrocodone + Misoprostol | 1 (0.0%) | - | 1 (0.3%) | - | - |
| | Acetaminophen/Hydrocodone + Oxycodone + Ketorolac | 1 (0.0%) | 1 (0.0%) | - | - | - |
| | Acetaminophen/Oxycodone + Fentanyl + Ketorolac | 1 (0.0%) | - | 1 (0.3%) | - | - |
| | Acetaminophen/Oxycodone + Hydromorphone + Oxycodone | 1 (0.0%) | 1 (0.0%) | - | - | - |
| | Acetaminophen/Oxycodone + Ketorolac + Ibuprofen | 1 (0.0%) | 1 (0.0%) | - | - | - |
| | Acetaminophen/Oxycodone + Oxycodone + Ketorolac | 1 (0.0%) | 1 (0.0%) | - | - | - |
| | Buprenorphine/Naloxone + Ibuprofen | 1 (0.0%) | 1 (0.0%) | - | - | - |
| | Codeine + Ketorolac + Ibuprofen | 1 (0.0%) | 1 (0.0%) | - | - | - |

**Table 3.** (Continued)

| Medication Group | Medications or Medication Combinations | Overall (N = 28717) | Gynecology Clinic (n = 25412) | OR/ General Surgery (n = 319) | Primary Care Clinic (n = 552) | Comprehensive Women's and Gender Diverse Primary Care Clinic (n = 2434) |
|---|---|---|---|---|---|---|
| | Codeine + Misoprostol | 1 (0.0%) | 1 (0.0%) | - | - | - |
| | Codeine + Morphine + Ibuprofen | 1 (0.0%) | 1 (0.0%) | - | - | - |
| | Fentanyl + Hydromorphone + Ibuprofen | 1 (0.0%) | 1 (0.0%) | - | - | - |
| | Fentanyl + Hydromorphone + Meperidine + Oxycodone | 1 (0.0%) | 1 (0.0%) | - | - | - |
| | Fentanyl + Hydromorphone + Oxycodone + Cervical LP Cream | 1 (0.0%) | - | 1 (0.3%) | - | - |
| | Fentanyl + Hydromorphone + Oxycodone + Misoprostol | 1 (0.0%) | - | 1 (0.3%) | - | - |
| | Fentanyl + Ketorolac | 1 (0.0%) | 1 (0.0%) | - | - | - |
| | Fentanyl + Misoprostol + Cervical Lidocaine Gel | 1 (0.0%) | 1 (0.0%) | - | - | - |
| | Fentanyl + Oxycodone + Ketorolac + Ibuprofen | 1 (0.0%) | 1 (0.0%) | - | - | - |
| | Hydromorphone + Meperidine + Ibuprofen | 1 (0.0%) | 1 (0.0%) | - | - | - |
| | Hydromorphone + Meperidine + Ketorolac | 1 (0.0%) | - | 1 (0.3%) | - | - |
| | Hydromorphone + Meperidine + Misoprostol | 1 (0.0%) | - | 1 (0.3%) | - | - |
| | Hydromorphone + Oxycodone + Ketorolac + Ibuprofen | 1 (0.0%) | - | 1 (0.3%) | - | - |
| | Hydromorphone + Oxycodone + Meperidine + Misoprostol | 1 (0.0%) | - | 1 (0.3%) | - | - |
| | Ketorolac + Cervical LP Cream | 1 (0.0%) | 1 (0.0%) | - | - | - |
| | Ketorolac + Misoprostol + Paracervical Block | 1 (0.0%) | 1 (0.0%) | - | - | - |
| | Oxycodone + Hydromorphone + Morphine + Meperidine + Misoprostol + Ibuprofen | 1 (0.0%) | - | 1 (0.3%) | - | - |
| | Oxycodone + Ketorolac + Naproxen | 1 (0.0%) | 1 (0.0%) | - | - | - |
| | Oxycodone + Misoprostol | 1 (0.0%) | 1 (0.0%) | - | - | - |
| | Oxycodone + Misoprostol + Ketorolac | 1 (0.0%) | - | 1 (0.3%) | - | - |
| | Paracervical Block + Ibuprofen | 1 (0.0%) | 1 (0.0%) | - | - | - |
| | Tramadol + Ketorolac | 1 (0.0%) | 1 (0.0%) | - | - | - |
| | Tramadol + Ketorolac + Naproxen | 1 (0.0%) | 1 (0.0%) | - | - | - |
| | Tramadol + Misoprostol + Ibuprofen | 1 (0.0%) | 1 (0.0%) | - | - | - |
| | Tramadol + Naproxen + Ibuprofen | 1 (0.0%) | - | - | - | 1 (0.0%) |

NSAIDs = Non-Steroidal Anti-inflammatory Drugs; WH = Women's Health; PC = Primary Care; OR = Operating Room.

[1]NSAIDs included oral naproxen, intramuscular or oral ketorolac, and oral ibuprofen, or any combination. It is important to note that taking a combination of NSAIDs at the same time is not recommended, and these medications were prescribed within a 24-hour window of the procedure.

[2] Opioid analgesics included acetaminophen/hydrocodone, oxycodone, acetaminophen/oxycodone, hydromorphone, tramadol, acetaminophen/codeine, meperidine, morphine, fentanyl, buprenorphine/naloxone, codeine, or any combination.

[3] Prostaglandins included the cervical ripening agents such as misoprostol and dinoprostone.

[4]Lidocaine included cervical lidocaine gel, cervical lidocaine spray, intrauterine lidocaine, cervical lidocaine-prilocaine (LP) cream, paracervical blocks (also known as the lidocaine block), and diclofenac plus cervical lidocaine gel.

[5]Combination or Other included any combinations of two or more of the previously mentioned groups (i.e., misoprostol with ibuprofen).

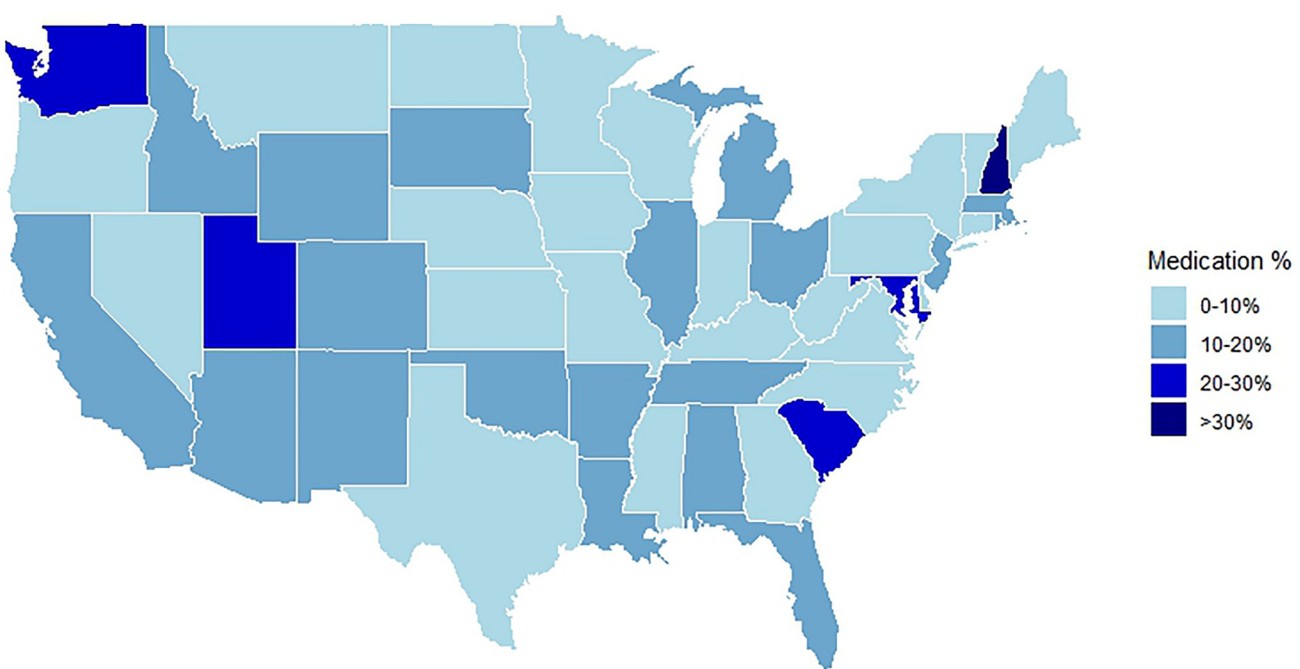

**Fig 2. Percentage among the 24,010 IUD insertions performed within VHA prescribed pain medication across the United States, January 1st, 2018, to October 13th, 2023. F2L:** The blue scale gradient on the United States map above depicts the average proportion of intrauterine device insertion procedures where any pain medication was prescribed within the Veterans Affairs Healthcare System from January 1st, 2018, to October 13th, 2023, with the darker blues associated with a higher proportion of procedures with prescribed medications. Exact percentages can be found in S2 Appendix.

## Geographic variation in pain treatment

Our analysis revealed regional variations throughout the US regarding the prescription of pain medication for IUD insertions, with the proportion of those prescribed any form of pain medication varying from 9.0% in the Northeast to 12.1% in the West (**Table 2**). However, a closer examination at the state level uncovered more significant differences, with the percentage of procedures involving any form of pain medication ranging from 1.0% in Missouri to 32.0% in New Hampshire (**Fig 2, and S4 Appendix**). Furthermore, our results indicate that the prescription of pain medications other than ibuprofen for procedures across the US states was notably low, varying from 0.6% in Missouri to 26.7% in the District of Columbia.

## Discussion

This assessment provides new insights into the current state of prescribed pain management for Veterans undergoing IUD insertions within the VHA, addressing gaps in the literature regarding the adequacy of pain interventions in this population. Although pain associated with IUD insertion is well documented, our assessment revealed that only 11.4% of the 28,717 IUD insertions procedures performed at VHA, nationally, during the assessment period, had any form of documented pain medication, highlighting the opportunity to improve patient care. Despite an average annual increase of 0.52% in the proportion of those prescribed pain management between January 1st, 2018, to October 13th, 2023, this slow progression in the adoption of pain management strategies highlights the need for more systemic change in clinical guidance to ensure that effective pain relief is consistently offered.

We observed significant variation between US states in the proportion of patients who received any form of pain medication, ranging from 1.0% in Missouri to 32.0% in New Hampshire, even though the majority of the IUD insertion procedures were performed within high complexity facilities (91.7%) with similar available resources. Depending on location, only 0.6% to 26.7% of procedures were prescribed analgesics other than ibuprofen. While our data could not assess the potential causes of the variation, possible factors include regional practices, provider training, patient preferences, and/or specific protocols within each VHA facility. The absence of a standardized operating procedure or mandate for pain management may contribute to these inconsistencies. Bias, including both unconscious bias and gender-related biases, may also play a role in prescribing practices, as a 2018 meta-analysis on gender bias in healthcare found that women's pain is often under-treated in healthcare settings [38].

Despite multiple studies concluding ibuprofen is ineffective at reducing pain during or after IUD insertion [3, 39, 40], we found it to be the most frequently prescribed pain intervention. Previous research has shown that naproxen is the only NSAID shown to be effective in decreasing pain during IUD insertion [41, 42]. Interestingly, our results show naproxen, as a stand-alone treatment, was only prescribed 0.5% of the time. Other options, like prostaglandins, which are often used to induce cervical ripening in vaginal delivery, were at one time thought to be effective in helping ease IUD placement and, therefore, any associated pain [42]. However, more recently published results have found prostaglandins to be ineffective in easing the IUD insertion process and associated pain reduction [43, 44]. Only 1.6% of procedures captured in this assessment were prescribed prostaglandins.

A small number of prescribed opioid analgesics were prescribed, potentially for patients with a more complicated health history than our data was unable to uncover (i.e., previously embedded IUD, scarring, etc.). To our knowledge, most opioids have not yet been tested for their effectiveness in reducing pain during or after an IUD procedure with the exception of tramadol, which has been shown to be effective in reducing pain with IUD placement [35, 45]. In this assessment, tramadol was prescribed in less than 1% of procedures, often in combination with other prescribed pain interventions.

Various formulations of lidocaine have been shown to be effective in reducing pain associated with IUD insertions. For example, cervical lidocaine 2% gel (either with or without the combination of diclofenac) has shown some effectiveness for reducing pain associated with IUD placement [46, 47]. Cervical lidocaine 4% gel, LP (lidocaine; prilocaine) cream, and the paracervical block with lidocaine have all been shown to be effective either with tenaculum placement or after the procedure [14, 46]. LP cream alone has also been shown to reduce pain of tenaculum placement by 24% and pain from IUD insertion by 28% [46]. Yet despite this evidence, only 0.2% of the procedures captured in our sample were prescribed any form lidocaine. Additionally, paracervical blocks have been shown to be effective in reducing overall pain associated with this procedure [48]. However, we found only 6 occurrences (0.0%) of the paracervical block being prescribed as a standalone pain intervention for IUD insertions within our sample, and only 8 occurrences within the dataset in its entirety.

Some patients may experience more pain or discomfort than others during IUD insertion depending on their medical history. Previous studies have shown that patients with a history of chronic pelvic pain, sexual pain, dysmenorrhea, or painful periods, as well as those who are post-partum, postmenopausal, have a history of sexual trauma, and those who suffer from anxiety, are more likely to experience greater pain with IUD insertions [34]. We found 62.7% of our cohort were diagnosed with anxiety disorders, 38.2% had a history of MST, and 22.6% had a history of chronic pelvic pain. Additionally, nulliparous patients have been found to report significantly higher pain when undergoing IUD insertions compared to parous patients [49], which comprised 94.4% of our sample.

Notably, we observed significant between-group differences for several patient characteristics. Patients with a history of MST, anxiety disorders, dysmenorrhea, dyspareunia, or chronic pelvic pain were more likely to be prescribed pain medication than those without these conditions. Similarly, we found differences based on rurality, with 11.9% of urban patients receiving pain medication compared to 9.4% of rural patients. Racial and socioeconomic disparities in pain management, particularly in gynecological care, are well-documented, with historical inequities affecting the treatment of Black women [50]. In our assessment, we found slightly higher proportions of Black (12.8%) and Native Hawaiian or Pacific Islander (13.2%) patients prescribed pain medication compared to Non-Hispanic Whites (10.6%) and American Indian or Alaskan Natives (8.3%). Despite these differences, our data demonstrates a significant lack of prescribed pain interventions for IUD insertions, and when initiated, VHA providers were disproportionally prescribing medications that have not been shown to be effective in reducing pain during or after the IUD insertion. Furthermore, there was a significant geographic variation in pain treatment practices, which suggests a lack of a comprehensively implemented strategy across the VHA.

Several countries and organizations have systematically taken steps to address analgesic options for pain related to IUD placements, and understanding these efforts may provide policy insights. For example, in 2020, clinical guidelines with recommended pain interventions were adopted in Canada [46] and reflected in an official statement by the Society of Obstetricians and Gynecologists of Canada in 2023 [51]. These guidelines strongly recommend that healthcare providers counsel patients on the various pain management options available before the insertion procedure, taking into account the patient's medical history and the specific techniques used, as these factors can influence the effectiveness of the pain intervention. Similarly, in 2024, the CDC updated their "US Selected Practice Recommendation for Contraceptive Use" following a review of scientific evidence and consultation with national experts [21]. These updated recommendations include guidance on the provision of medication for IUD placement, as well as management of bleeding irregularities during implant use. Additionally, a public petition in the United Kingdom amassed over 35,000 signatures advocating for better pain relief for IUD insertions [52], leading the Faculty of Sexual and Reproductive Healthcare and the Royal College of Obstetricians and Gynecologists to release a statement updating their guidance to members; with the president, Dr Edward Morris, stating, "...*We believe that unbearable pain during any gynecological procedure is unacceptable and all specialists working in women's health; specialist nurses, GPs and gynecologists need to listen and take account of what is being said.*" [53].

There could be several reasons for the lack of an apparent comprehensive approach in our population. For example, one reported potential reason could be that some providers perceive pain to be less than the patient experiences during gynecological procedures. A 2015 study found that the mean patient rated maximum pain during the IUD insertion was 64.8 millimeters (mm) on a 100 mm visual analogue scale (VAS; where 0 mm represents "no pain" and 100 mm represents "the worst imaginable pain") compared to only 35.3 mm when rated by the provider [54]. In addition, the previously referenced 2018 meta-analysis on gender bias in healthcare suggests that the lack of pain medication may be due to a healthcare system's failure to legitimize women's pain and revealed that a women's pain is more likely to be described as "hysterical" or "sensitive." [38]. A 2020 VHA study further highlighted these perceptions, with some female Veterans reporting that VA specialty care providers did not take their symptoms seriously, attributing their health concerns to hormonal fluctuations [55]. By initiating standardized and data informed clinical guidelines for providers to counsel patients on what to expect during IUD placement, including effective pain control options, VHA would not only

be providing better care, but would also strengthen the trust between Veterans and their providers.

Our VA population may face unique Veteran-related health challenges, which may limit the generalizability of our assessment to non-VA settings. This assessment is limited as it is a retrospective analysis of routinely collected clinical data demonstrating population level associations rather than causations. However, this retrospective design also provides valuable real-world insights about nationwide clinical trends and practices that may be challenging to achieve in a prospective trial due to enrollment size and the Hawthorne effect [56]. Although we queried both structured and unstructured data fields to capture all analgesic options, we may have missed medications purchased over the counter (OTC) or prescribed outside of the VA. Despite this, because VA practices purposeful adverse patient selection, many Veterans rely on VA for OTC medications, offering a broader scope of care and a more complete record of outpatient analgesic use. However, Veterans with better financial means may not rely on VA for OTC medications, and it is possible that some of these patients did not report their pain medication use to their clinicians, even though medication reconciliation is a national VA system-wide policy for clinical visits, and the VA EHR database is designed to include non-VA prescribed OTC medications. Furthermore, our method of capturing any non-opioid pain medication within 45 days of IUD placement was designed to comprehensively include any potential pain medication associated with the procedure. Although this method increased the sensitivity for detecting prescribed pain medications, it reduced the specificity that the medications prescribed during this time were intended for IUD associated pain. As a result, the rates of pain medication prescription for IUD placement may actually be even lower than we reported, further emphasizing the care gap and need for informed action.

It is important to note that evidence regarding pain levels among individuals receiving IUDs varies considerably across studies, likely due to differences in pain measurement methods, patient characteristics, and sample sizes. Some studies report moderate to severe pain with mean scores of 64.8 mm on a 100 mm VAS [38], while others show lower pain levels, with mean scores ranging from 1.0 cm on a 10 cm VAS [3] to 38.0 mm on a 100 mm VAS [39] during insertion. This variability underscores the need for consistent pain assessment methods and highlights that pain perception can differ significantly across individuals. Unfortunately, we were unable to capture patient-reported pain scores for the majority of IUD insertions in this assessment, as pain was not consistently documented before, during, or after the procedure. Additionally, for the few with documented scores, due to the methods of documentation, we were unable to ascertain at what time the pain score was recorded, which is critical when assessing perceived pain at the time of insertion. This limitation prevented us from directly analyzing the relationship between reported pain and the use of pain management interventions. Given the importance of understanding patients' pain experiences during this procedure, we recommend that routine collection of pain levels at the time of IUD insertion be integrated into clinical practice.

A notable strength of this assessment is the large population-based sample of Veterans, from across the US, utilizing documented clinical data (e.g., diagnoses, medication use, and demographics). In addition, our approach leveraged both structured and unstructured data sources from both inpatient and outpatient sources, to provide a comprehensive assessment of pain prescriptions. Future steps could include qualitative interviews with key stakeholders (such as Veterans, providers, and administrators) to better understand the decision-making processes surrounding the use of IUDs, as well as the patient-provider communications that occur regarding pain management. Understanding the perspectives of these stakeholders will be crucial in developing strategies that align with both patient needs and provider capabilities. Future work could also incorporate the routine collection and analysis of patient-reported

pain scores during IUD insertion. This would allow for a more direct assessment of pain levels in relation to prescribed interventions and provide valuable insights into optimizing pain management strategies for our specific patient population. Additionally, further exploration of differences in pain management between male and female Veterans, as some research indicates variations in methods and perceived biases [55, 57], could help address any gender-related disparities in care.

Given the recent changes surrounding reproductive rights, such as the overturning of Row v. Wade [58], there is increasing attention on the accessibility of contraceptives, particularly in states where abortion access is restricted or banned. This may influence both patient preferences and provider recommendations for IUDs as a highly effective and long-term method to prevent unintended pregnancies. Although our assessment does not specifically assess the impact of these legal changes, future research could explore how shifts in reproductive healthcare access affect the utilization of IUDs and other forms of contraception within the VHA and nationally. Analyzing potential changes after data-informed policies are implemented, as well as understanding patient characteristics that are predictive of being prescribed pain interventions, will be of additional value.

The aim of this assessment was to evaluate the current state of prescribed pain intervention for our Veteran population undergoing IUD insertion procedures at a national level. This evaluation is a crucial step towards improving pain management practices. Our findings allow for data-informed development and adoption of evidence-based guidelines and the standardization of pain management protocols within the VHA. Enhancing provider education is paramount to ensure healthcare professionals are well-informed about effective pain control options. It is essential that top-down systemic recommendations or requirements, mandated by leadership, be implemented to ensure that providers consistently discuss pain management options with patients prior to IUD insertion procedures. Similarly to how the VHA has successfully rolled out training for other procedures, such as empirically supported therapies [59], these organization-wide policies would standardize care and improve alignment with evidence-based practices. Pre-procedural discussions would ensure that patients are informed of their pain management options and reduce disparities in treatment. Additionally, promoting patient-centered care models that respects the unique needs of Veterans who are AFAB and integrating patient feedback into care processes will be essential. These measures will ensure that pain management during IUD insertion procedures is both effective and tailored to individual patient needs.

## Conclusion

Although there has been a steady increase in the use of prescribed pain treatment for IUD procedures in the VHA, the overall rate of prescribed pain interventions for IUD procedures remains low. Furthermore, the most commonly prescribed analgesic medications for IUD insertions in VHA are different than what the current literature has shown to be effective treatments. The intent of this assessment is that it will contribute to data informed interventions, policies, education, and processes that promote optimal standardized pain medications for patients undergoing IUD insertion procedures within the VHA.

## Supporting information

**S1 Appendix. Individual medications included in each medication group (Non-steroidal anti-inflammatory drugs, opioid analgesics, prostaglandins, lidocaine, combination or other).**
(DOCX)

**S2 Appendix. Intrauterine device insertion procedure flowchart.**
(DOCX)

**S3 Appendix. Patient characteristics among the 24,010 IUD insertions performed within VHA between January 1st, 2018, to October 13th, 2023, with documented pain medication for their intrauterine device insertion procedure by medication type compared to those without any prescribed pain medication.**
(DOCX)

**S4 Appendix. Proportion of United States women Veterans prescribed pain medication for IUD procedures by state, 2018–2023.**
(DOCX)

## Acknowledgments

**Disclaimer:** The findings and conclusions in this report are those of the authors and do not necessarily represent the views of the U.S. Department of Veterans Affairs or the United States Government.

## Author Contributions

**Conceptualization:** Anna D. Ware, Zach P. Veigulis.

**Data curation:** Anna D. Ware, Terri L. Blumke.

**Formal analysis:** Anna D. Ware.

**Investigation:** Anna D. Ware.

**Methodology:** Anna D. Ware.

**Project administration:** Anna D. Ware.

**Supervision:** Thomas F. Osborne.

**Validation:** Anna D. Ware, Terri L. Blumke, Peter J. Hoover, Jacqueline M. Ferguson, Malvika Pillai.

**Visualization:** Anna D. Ware, Terri L. Blumke, Peter J. Hoover, Jacqueline M. Ferguson, Malvika Pillai, Thomas F. Osborne.

**Writing – original draft:** Anna D. Ware, Terri L. Blumke, Peter J. Hoover, Zach P. Veigulis, Jacqueline M. Ferguson, Malvika Pillai, Thomas F. Osborne.

**Writing – review & editing:** Anna D. Ware, Terri L. Blumke, Peter J. Hoover, Zach P. Veigulis, Jacqueline M. Ferguson, Malvika Pillai, Thomas F. Osborne.

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
