## [Decision Letter · Decision Letter 0]

23 Sep 2024

PONE-D-24-29921National Assessment on the Frequency of Pain Medication Prescribed for Intrauterine Device Insertion Procedures within the Veterans Affairs Health Care SystemPLOS ONE

Dear Dr. Ware,

Thank you for submitting your manuscript to PLOS ONE. After careful consideration, we feel that it has merit but does not fully meet PLOS ONE’s publication criteria as it currently stands. Therefore, we invite you to submit a revised version of the manuscript that addresses the points raised during the review process.

Specifically, the reviewers have raised concerns about the introduction’s discussion of pain levels experienced during IUD insertion, highlighting the need for a more critical review of existing studies and the potential discrepancies in pain measurements across various populations. Additionally, they noted that your study did not include patient-reported pain data from the VA EHR system, which could limit the conclusions on pain management disparities. This should be addressed as a limitation in the study, with a potential recommendation for the routine collection of this data in future studies. Clarifying the rationale behind covariates used, addressing the apparent discrepancy in IUD insertion rates among the Veteran population, and harmonizing the results tables for clarity are also key areas requiring attention. The reviewers’ comments are listed below, and we look forward to receiving a revised manuscript that reflects these important changes.

Please submit your revised manuscript by Nov 07 2024 11:59PM.  If you will need more time than this to complete your revisions, please reply to this message or contact the journal office at plosone@plos.org. Please include the following items when submitting your revised manuscript:

We look forward to receiving your revised manuscript.

Kind regards,

Prita Abhay Dhaimade

Academic Editor

PLOS ONE

2. We note that you have indicated that there are restrictions to data sharing for this study. For studies involving human research participant data or other sensitive data, we encourage authors to share de-identified or anonymized data. However, when data cannot be publicly shared for ethical reasons, we allow authors to make their data sets available upon request. For information on unacceptable data access restrictions, please see http://journals.plos.org/plosone/s/data-availability#loc-unacceptable-data-access-restrictions. Before we proceed with your manuscript, please address the following prompts: a) If there are ethical or legal restrictions on sharing a de-identified data set, please explain them in detail (e.g., data contain potentially identifying or sensitive patient information, data are owned by a third-party organization, etc.) and who has imposed them (e.g., a Research Ethics Committee or Institutional Review Board, etc.). Please also provide contact information for a data access committee, ethics committee, or other institutional body to which data requests may be sent. b) If there are no restrictions, please upload the minimal anonymized data set necessary to replicate your study findings to a stable, public repository and provide us with the relevant URLs, DOIs, or accession numbers. Please see http://www.bmj.com/content/340/bmj.c181.long for guidelines on how to de-identify and prepare clinical data for publication. For a list of recommended repositories, please see https://journals.plos.org/plosone/s/recommended-repositories. You also have the option of uploading the data as Supporting Information files, but we would recommend depositing data directly to a data repository if possible. Please update your Data Availability statement in the submission form accordingly.

Additional Editor Comments (if provided):

Reviewers' comments:

Reviewer's Responses to Questions

**Comments to the Author**

1. Is the manuscript technically sound, and do the data support the conclusions?

Reviewer #1: Yes

Reviewer #2: Yes

2. Has the statistical analysis been performed appropriately and rigorously? 

Reviewer #1: Yes

Reviewer #2: Yes

3. Have the authors made all data underlying the findings in their manuscript fully available?

Reviewer #1: No

Reviewer #2: No

4. Is the manuscript presented in an intelligible fashion and written in standard English?

Reviewer #1: Yes

Reviewer #2: Yes

5. Review Comments to the Author

Reviewer #1: Thanks for the opportunity to review this manuscript. The topic is extremely important and addresses a gap in the literature surrounding healthcare for women Veterans in the form of IUD and associated pain management. Contraceptive care for women is understudied, pain during IUD insertion is understudied, and each of these topics among Veterans specifically is remarkably understudied. Thanks to the authors for contributing to this area of work!

A few questions arose during my review that I have numbered below:

1. Is there other relevant work regarding pain medications among Veterans, and in particular, disparities between men and women Veterans that would be helpful to report in the introduction? While the IUD is a procedure that occurs in individuals assigned female sex at birth, the disparity in pain management between male and female Veterans is relevant here.

2. The authors might give a bit more attention to what effective pain management strategies are and are not in the introduction for context that would help to interpret later findings.

3. Authors might note why IUD is preferable or the only option for some individuals when considering modality of contraception.

4. The authors note that Veterans are more likely to have IUD than non-Veterans. Is there any work on why this difference exists? Is there evidence that VHA providers are more likely to advise IUD vs. oral contraception or other methods? This is particularly interesting in light of the high rates of MST among women Veterans and research demonstrating that some individuals with sexual trauma may avoid internal exams, pap smears, or other aspects of gynecological care.

5. Are there other disparities to be mindful to acknowledge, such as racial or ethnic disparities, socioeconomic disparities? There is significant historical context around gynecological care for Black women, for example, that may impact treatment of pain as well as the actual physician recommendations.

6. It’s likely beyond the scope of this study, but it would be interesting to comment on the impact of Roe v Wade being overturned as related to contraception recommendations (particularly in states where abortion is banned).

7. The early sentences of the Discussion section largely mirror the early sentences in the introduction; consider reframing discussion section around how your manuscript addresses gaps in the literature.

8. I might note that the “steady increase” in medications prescribed is relatively unimpressive given that IUD insertion pain is not new information. This is in no way meant as a critique of the authors, but a suggestion to make your statements stronger to pull for systemic change in guidance.

9. The authors suggest reasons for variation in prescribing practices; I would also include bias.

10. It looks like between-group differences for MST, rurality, anxiety disorders, and chronic pelvic pain were examined but only noted in the table. It may be worth pointing readers to this in the text, as these are interesting findings that, without close review of the table, may be missed.

11. In the recommendations, it may be important to emphasize a top-down recommendation or requirement that providers have these medication discussions prior to the procedure. Similar to how VHA providers are trained in other procedures (i.e., empirically supported therapy rollouts), it seems urgent that providers are told, not asked, what to do to better align with best practice.

12. Did the authors distinguish between pain medications prescribed within 1 day of the insertion procedure and those within 44 days? This may help speak to the last issue mentioned in the limitations section, where authors note that these rates may be underestimates of the true rates.

13. Future steps may include qualitative interviews with key stakeholders, including Veterans, providers, administrators, etc., to better understand the choices, as well as the patient-provider communications that occur surrounding the choice to use an IUD and the associated pain.

Reviewer #2: This study examined frequency and type of pain medication treatment for intrauterine device (IUD) insertion procedures within the Veterans Health Administration. The authors reported that 11.4 percent of procedures had any form of prescribed pain medication identified, with ibuprofen being the most common type of pain medication even though it is not effective. This well-done study is novel in its use of large electronic health records (EHR) data to assess prevalence of pain medication treatment at the population level, trends in pain medication treatment over time, and differences in pain medication treatment by patient, facility and provider characteristics.

Major comment:

The authors need to describe more carefully and critically evidence regarding pain among people receiving IUDs, including how pain is measured. Their main evidence is a study according to which 78 percent of respondents reported moderate to severe pain levels, but that study only had 109 participants and focused on a specific population (nulliparous women aged 18-30; see reference number 5). Another study they mentioned (reference number 3) seemed to find a much lower level of reported pain (a median level of pain of 1.0cm on a 10cm visual analog scale), whereas a third study (reference number 47) reported a mean patient maximum pain of 6.48cm on the same scale. Moreover, studies on the effectiveness of pain medication report pain levels for the control group and thus provide further evidence (see for instance Table 2 in https://doi.org/10.1016/j.contraception.2011.10.015; Table 2 in https://doi.org/10.1016/j.contraception.2014.11.012; and Table I in https://doi.org/10.1016/j.ajog.2006.08.022). It seems as if the reported level of pain varies considerably across studies, which may be due to pain measurement, patient characteristics, or random sample variation. A more thorough description of reported pain among people receiving IUDs is important because the main take-away of this study was that levels of prescribed pain medication was low compared to the pain patients experienced.

Related to this point, it seems as the VA EHR data do not include reported pain of patients as part of IUD insertion procedures. This type of information would be very helpful to be able to assess overall pain levels in this population, and to examine disparities between experienced pain and pain medication treatment. The authors should clarify this aspect as a limitation of their study. They could also consider adding routine collection of pain levels as a recommendation.

Other comments:

1. Materials and Methods. It was not clear to me why the authors included some of the covariates, especially those related to facilities (e.g., facility complexity). The authors should clarify their motivation for including these covariates. They should also add this aspect of their analysis in the last paragraph summarizing their study.

2. Results. I was surprised by the small fraction of IUD insertion procedures relative to the number of adults patients who were assigned female at birth (28,717 of 1,614,650, or 1.8 percent), given that 23 percent of Veterans assigned female at birth and ages 18-45 reported use of some long-active reversible contraception (see introduction, p.4). The authors should explain this discrepancy.

3. Results. Table 1 shows prevalences of no pain prescription and *any* pain medication prescription by patient characteristics, whereas Table 2 shows prevalences of no pain prescription vs. *type* of pain prescription by facility and procedure characteristics. It would be helpful to harmonize this displays.

6. PLOS authors have the option to publish the peer review history of their article (what does this mean?). If published, this will include your full peer review and any attached files.

Reviewer #1: **Yes: **Emily B. K. Thomas

Reviewer #2: No

---

## [Author Response · Author response to Decision Letter 0]

16 Oct 2024

PLOS ONE

1265 Battery Street Suite 200

San Francisco, CA 94111

September 23, 2024

RE: Manuscript Number: PONE-D-24-29921

Dear Editors,

Thank you for the opportunity to revise our manuscript entitled, “National Assessment on the Frequency of Pain Medication Prescribed for Intrauterine Device Insertion Procedures within the Veterans Affairs Health Care System” for publication in PLOS ONE. We sincerely appreciate the reviewers’ feedback and have made considerable efforts to address all comments. Please see below the point-by-point response to the reviewers. 

Reviewer 1: Thanks for the opportunity to review this manuscript. The topic is extremely important and addresses a gap in the literature surrounding healthcare for women Veterans in the form of IUD and associated pain management. Contraceptive care for women is understudied, pain during IUD insertion is understudied, and each of these topics among Veterans specifically is remarkably understudied. Thanks to the authors for contributing to this area of work!

Response: Thank you for your kind assessment of our work. We greatly appreciate your thoughtful and thorough suggestions as we aim to address this important knowledge gap in scientific literature.

Reviewer 1: Is there other relevant work regarding pain medications among Veterans, and in particular, disparities between men and women Veterans that would be helpful to report in the introduction? While the IUD is a procedure that occurs in individuals assigned female sex at birth, the disparity in pain management between male and female Veterans is relevant here.

Response: Thank you for your insightful comment. Recent literature among Veterans does suggest some differences in pain management between male and female Veterans, although the findings are nuanced. For instance, Driscoll et al. (2017; https://doi.org/10.1093/pm/pnx023) found no significant differences in pain management methodologies between male and female Veterans. However, Murphy et al. (2016; https://www.rehab.research.va.gov/jour/2016/531/pdf/jrrd-2014-10-0250.pdf) found that while women reported shorter pain duration and were more likely to experience head or limb pain, they were initially prescribed fewer opioids compared to men. Despite this, opioid use did not differ significantly at follow-up. Additionally, Oliva et al. (2015; https://doi.org/10.1111/pme.12501) noted that women Veterans prescribed pain medications were more likely to receive guideline-recommended practices than their male counterparts. 

Further evidence of perceived bias among female Veterans indicates that gender-based assumptions may also influence care. For example, Mattocks et al. (2020; https://doi.org/10.1016/j.whi.2019.10.003) found that some female Veterans felt their symptoms were not taken seriously by VA specialty care providers, attributing this to gender-based assumptions, such as being told that their symptoms were due to hormonal fluctuations. These findings underscore that, while explicit disparities in pain management methodologies may not be significant, perceptions of gender bias in care remain an critical issue. 

Given the complexity and sometimes conflicting results of these studies, expanding the introduction to cover gender-based differences in pain management may divert the paper’s focus. Instead, we have chosen to address the issue of perceived gender bias within our discussion section. To this end, we have added the following sentences: 

“A 2020 VHA study further highlighted these perceptions, with some female Veterans reporting that VA specialty care providers did not take their symptoms seriously, attributing their health concerns to hormonal fluctuations. By initiating standardized and data informed clinical guidelines for providers to counsel patients on what to expect during IUD placement, including effective pain control options, we believe that VHA will not only provide better care, but also strengthen the trust between Veterans and their providers.”

Additionally, we have expanded the discussion to suggest that further exploration of pain management differences between male and female Veterans is warranted, as some research highlights potential variations in pain management methods and perceived biases. The added sentence is: 

“Additionally, further exploration of pain management differences between male and female Veterans, as some research indicates variations in methods and perceived biases, could help address any gender-related disparities in care.” 

In this expanded section of the discussion, we reference both Mattocks et al. (2020) and Olivia et al. (2015) to emphasize the importance of addressing potential biases and ensuring equitable pain management for all Veterans.

Reviewer 1: The authors might give a bit more attention to what effective pain management strategies is and are not in the introduction for context that would help to interpret later findings.

Response: Thank you for your suggestion. We agree that being more specific on which pharmaceuticals are effective in the introduction could be help with the interpretation of our later findings. We have added to the introduction to now state: 

“In attempts to mitigate pain during IUD insertion, several pharmacological interventions have been tested with mixed results. For example, certain formulations of lidocaine, such as cervical lidocaine gel and paracervical blocks, have shown effectiveness in reducing pain during the procedure. In contrast, most non‐steroidal anti‐inflammatory drugs (NSAIDs), including ibuprofen and ketorolac, have generally been ineffective at managing insertion-related pain, while naproxen has demonstrated some promise. Cervical ripening agents (prostaglandins), like misoprostol, were once thought to ease the insertion process, but recent studies have also cast doubt on their effectiveness. Additionally, while opioids are not commonly prescribed for this procedure, tramadol has been shown to reduce pain during IUD insertion in some studies.”

Reviewer 1: Authors might note why IUD is preferable or the only option for some individuals when considering modality of contraception.

Response: Thank you for your comment. We agree that this could be a value add to our introduction. Therefore, we have expanded on the first paragraph of the introduction to now state: 

“For some individuals, IUDs may be the preferable or only option for contraception due to specific health conditions. Those who cannot tolerate hormonal contraception, such as individuals with migraines, hypertension, or those at increased risk of thromboembolism, may opt for hormone-free copper IUDs. Additionally, hormonal IUDs also provide non-contraceptive benefits, such as reducing heavy menstrual bleeding and dysmenorrhea.”

Reviewer 1: The authors note that Veterans are more likely to have IUD than non-Veterans. Is there any work on why this difference exists? Is there evidence that VHA providers are more likely to advise IUD vs. oral contraception or other methods? This is particularly interesting considering the high rates of MST among women Veterans and research demonstrating that some individuals with sexual trauma may avoid internal exams, pap smears, or other aspects of gynecological care. 

Response: Thank you for your comment. The study we reference by Koenig et al. (2019), entitled “Factors associated with long-acting reversible contraception use among women Veterans in the ECUUN study,” found that increased use of LARC was not linked to provider- or facility-level factors. However, it did find that women receiving care VA women's health clinics more likely to receive LARCs. The author’s note that this “may suggest that requisite training of VA primary care providers as women's health providers, if practicing within a women's health clinics, along with a high target of women Veterans being seen by these women’s health providers allows providers, regardless of their location and characteristics, to provide comprehensive care for women.” While there is no direct evidence showing that VHA providers recommend IUDs over oral contraception, the structure of VA care – offering no or minimal cost for all contraceptive options – coupled with the focus on comprehensive primary care, ensures that Veterans have easier access to LARCs. This may contribute to the higher rates of LARC use, including IUDs, among Veterans compared to the general population, where cost and access barriers are more prevalent. Additionally, LARCs are often preferred for their long-term efficacy and convenience, which could be a factor for Veterans seeking reliable and low-maintenance contraception. 

To clarify this in our paper, we expanded the second to last paragraph of the introduction to now state: 

“To meet the needs of this rapidly expanding population, VHA has focused on key initiatives to enhance access to comprehensive women’s healthcare through specifically trained providers. Veterans can now access a full range of contraceptive options with no or minimal out-of-pocket expenses as part of their comprehensive primary care. This enhanced accessibility, coupled with the availability of specifically trained providers in VA women’s health clinics, enables Veterans to access LARCs more readily than the general population. Consequently, LARC utilization rates, including IUDs and implants, have been reported to be higher among Veterans than in the general population. According to a national telephone survey, approximately 23% of Veterans assigned female at birth (AFAB) of childbearing age (18-45 years) at risk for unintended pregnancy utilize a LARC, compared to only 11% of the general US population. This survey also found that 65% of these Veterans received their IUD within a VA outpatient clinic.” 

Reviewer 1: Are there other disparities to be mindful to acknowledge, such as racial or ethnic disparities, socioeconomic disparities? There is significant historical context around gynecological care for Black women, for example, that may impact treatment of pain as well as the actual physician recommendations. 

Response: Thank you for your comment. Racial and socioeconomic disparities in gynecological care have been well documented. We have added a sentence to our discussion section to acknowledge this disparity: 

“Racial and socioeconomic disparities in pain management, particularly in gynecological care, are well-documented, with historical inequities affecting the treatment of Black women. In our assessment, we found slightly higher proportions of Black (12.8%) and Native Hawaiian or Pacific Islander (13.2%) patients prescribed pain medication compared to Non-Hispanic Whites (10.6%) and American Indian or Alaskan Natives (8.3%).” 

We also added the following reference: Hoffman KM, Trawalter S, Axt JR, Oliver MN. Racial bias in pain assessment and treatment recommendations, and false beliefs about biological differences between blacks and whites. Proc Natl Acad Sci U S A. 2016;113(16):4296-4301. doi:10.1073/pnas.1516047113

Reviewer 1: It’s likely beyond the scope of this study, but it would be interesting to comment on the impact of Roe v Wade being overturned as related to contraception recommendations (particularly in states where abortion is banned). 

Response: Thank you for your insight regarding the impact of the overturning of Roe v. Wade on contraception recommendations. We agree that this is a critical issue and have added to the discussion acknowledging the potential influence of this legal change on contraceptive choices, particularly in states where abortion access is restricted or banned. While it is beyond the scope of our current assessment, we have highlighted that future research could explore how these shifts affect IUD use within the VA and nationally: 

“Given the recent legislative changes surrounding reproductive rights, such as the overturning of Row v. Wade, there is increasing attention on the accessibility of contraceptives, particularly in states where abortion access is restricted or banned. This may influence both patient preferences and provider recommendations for IUDs as a highly effective and long-term method to prevent unintended pregnancies. Although our assessment does not specifically assess the impact of these legal changes, future research could explore how shifts in reproductive healthcare access affect the utilization of IUDs and other forms of contraception within the VHA and nationally.”

Reviewer 1: The early sentences of the Discussion section largely mirror the early sentences in the introduction; consider reframing discussion section around how your manuscript addresses gaps in the literature.

Response: Thank you for your comment. We have revised this section of the discussion to center more on how our manuscript addresses gaps in literature: 

“This assessment provides new insights into the current state of prescribed pain management for Veterans undergoing IUD insertions within the VHA, addressing gaps in the literature regarding the adequacy of pain interventions in this population. Although pain associated with IUD insertion is well documented, our assessment revealed that only 11.4% of the 28,717 IUD insertions procedures performed at VHA, nationally, during the assessment period, had any form of documented pain medication, highlighting the opportunity to improve patient care.”

Reviewer 1: I might note that the “steady increase” in medications prescribed is relatively unimpressive given that IUD insertion pain is not new information. This is in no way meant as a critique of the authors, but a suggestion to make your statements stronger to pull for systemic change in guidance. The authors suggest reasons for variation in prescribing practices; I would also include bias.

Response: Thank you for your thoughtful comment. We agree that the wording of “steady increase” does not adequately reflect the fact that it remains a very slow adoption rate of pain interventions offered for this procedure. We have revised this section to now state: 

“Despite an average annual increase of 0.52% in the proportion of those prescribed pain management between January 1st, 2018, to October 13th, 2023, this slow progression in the adoption of pain management strategies highlights the need for more systemic change in clinical guidance to ensure that effective pain relief is consistently offered.”

Additionally, we have added a sentence to also suggest potential bias as a reason for variation in prescribing practices: 

“While our data could not assess the potential causes of the variation, possible factors include regional practices, provider training, patient preferences, and/or specific protocols within each VHA facility. The absence of a standardized operating procedure or mandate for pain management may contribute to these inconsistencies. Bias, including both unconscious bias and gender-related biases, may also play a role in prescribing practices, as previous research has shown that women’s pain is often under-treated in healthcare settings.”

Reviewer 1: It looks like between-group differences for MST, rurality, anxiety disorders, and chronic pelvic pain were examined but only noted in the table. It may be worth pointing readers to this in the text, as these are interesting findings that, without close review of the table, may be missed.

Response: Thank you for your insight. We have added a sentence to the results section to discuss the rurality of patient’s resident results: 

“Additionally, 11.9% of patients with residing in an urban location were prescribed pain medication compared to only 9.4% of patients residing in a rural location (p=0.003; Table 1).” 

We also added a sentence to the discussion to direct the readers to these findings: 

“Notably, we observed significant between-group differences for several patient characteristics. Patients with a history of MST, anxiety disorders, dysmenorrhea, dyspareunia, or chronic pelvic pain were more likely to be prescribed pain medication than those without these conditions. Similarly, we found differences based on rurality, with 11.9% of urban patients receiving pain medication compared to 9.4% of rural patients.”

Reviewer 1: In the recommendations, it may be important to emphasiz

---

## [Decision Letter · Decision Letter 1]

9 Dec 2024

National Assessment on the Frequency of Pain Medication Prescribed for Intrauterine Device Insertion Procedures within the Veterans Affairs Health Care System

PONE-D-24-29921R1

Dear Anna Denee’ Ware

We’re pleased to inform you that your manuscript has been judged scientifically suitable for publication and will be formally accepted for publication once it meets all outstanding technical requirements.

Kind regards,

Prita Abhay Dhaimade

Academic Editor

PLOS ONE

Additional Editor Comments (optional):

Reviewers' comments:

Reviewer's Responses to Questions

**Comments to the Author**

1. If the authors have adequately addressed your comments raised in a previous round of review and you feel that this manuscript is now acceptable for publication, you may indicate that here to bypass the “Comments to the Author” section, enter your conflict of interest statement in the “Confidential to Editor” section, and submit your "Accept" recommendation.

Reviewer #1: All comments have been addressed

Reviewer #2: All comments have been addressed

2. Is the manuscript technically sound, and do the data support the conclusions?

Reviewer #1: Yes

Reviewer #2: Yes

3. Has the statistical analysis been performed appropriately and rigorously? 

Reviewer #1: Yes

Reviewer #2: Yes

4. Have the authors made all data underlying the findings in their manuscript fully available?

Reviewer #1: No

Reviewer #2: No

5. Is the manuscript presented in an intelligible fashion and written in standard English?

Reviewer #1: Yes

Reviewer #2: Yes

6. Review Comments to the Author

Reviewer #1: The authors have sufficiently addressed the comments from both reviewers and revised the manuscript accordingly. Great work!

Reviewer #2: I appreciate the extensive revisions done by the authors. They have carefully addressed all my comments.

7. PLOS authors have the option to publish the peer review history of their article (what does this mean?). If published, this will include your full peer review and any attached files.

Reviewer #1: No

Reviewer #2: **Yes: **Stephan Lindner

---

## [Editor Report · Acceptance letter]

27 Dec 2024

PONE-D-24-29921R1 

PLOS ONE

Dear Dr. Ware, 

I'm pleased to inform you that your manuscript has been deemed suitable for publication in PLOS ONE. Congratulations! Your manuscript is now being handed over to our production team.

Kind regards, 

on behalf of

Dr. Prita Abhay Dhaimade 

Academic Editor

PLOS ONE